# Effect of Torrefaction on the Physiochemical Characteristics and Pyrolysis of the Corn Stalk

**DOI:** 10.3390/polym15204069

**Published:** 2023-10-12

**Authors:** Lei Chen, Xiangqian Chen, Yuxiao Zhao, Xinping Xie, Shuangxia Yang, Dongliang Hua, Chuanlei Wang, Tianjin Li

**Affiliations:** 1Shandong Provincial Key Laboratory of Biomass Gasification Technology, Energy Institute, Qilu University of Technology (Shandong Academy of Sciences), 19 Keyuan Road, Jinan 250014, China; chenl@sderi.cn (L.C.); chenxiangqian1998@163.com (X.C.); zhaoyx@sderi.cn (Y.Z.); xiexp@sderi.cn (X.X.); yangsx@sderi.cn (S.Y.); huadl@sderi.cn (D.H.); 2Jining Rencheng District Science and Technology Innovation Service Center, Jining 370811, China; 13789812782@163.com

**Keywords:** corn stalk, torrefaction, pyrolysis, gas products, morphology features

## Abstract

Torrefaction of biomass is one of the most promising pretreatment methods for deriving biofuels from biomass via thermochemical conversion processes. In this work, the changes in physicochemical properties and morphology features of the torrefied corn stalk, the changes in physicochemical properties and morphology features of the torrefied corn stalk were investigated. The results of this study showed that the elemental content and proximate analysis of the torrefied corn stalk significantly changed compared with those of the raw corn stalk. In particular, at 300 °C, the volatile content decreased to 41.79%, while the fixed carbon content and higher heating value increased to 42.22% and 21.31 MJ/kg, respectively. The H/C and O/C molar ratios of torrefied corn stalk at the 300 °C were drastically reduced to 0.99 and 0.27, respectively, which are similar to those of conventional coals in China. Numerous cracks and pores were observed in the sample surface of torrefied corn stalk at the torrefaction temperature range of 275 °C–300 °C, which could facilitate the potential application of the sample in the adsorption process and promote the release of gas products in pyrolysis. In the pyrolysis phase, the liquid products of the torrefied corn stalk decreased, but the H_2_/CO ratio and the lower heating value of the torrefied corn stalk increased compared with those of the raw corn stalk. This work paves a new strategy for the investigation of the effect of torrefaction on the physiochemical characteristics and pyrolysis of the corn stalk, highlighting the application potential in the conversion of biomass.

## 1. Introduction

Renewable energy resources play an important role in the improvement of sustainable energy systems around the world because continuous consumption of fossil fuels results in a series of global environmental issues, such as global warming and climate changes, which negatively impact human health [1,2]. Therefore, many nations are intensively dedicated to studies of renewable energy resources to alleviate the dependence on fossil fuels [3,4,5]. Lignocellulose biomass is a significant resource for the production of fuels due to various advantages of lignocellulose biomass, including zero carbon emission, extensive availability, improved stability, and safety [6,7,8]. However, raw lignocellulose biomass has the drawbacks of high moisture content, high oxygen content, and complex variations in chemical composition [8,9,10,11]. The torrefaction pretreatment is the most widely used and effective technology to overcome these drawbacks and convert biomass into more suitable biomass feedstock for pyrolysis [12,13,14,15].

The torrefaction process refers to a mild pre-pyrolysis process carried out at 200–300 °C under nitrogen, air, or carbon dioxide atmospheres only with the release of light volatiles and water [16,17,18,19,20,21]. Torrefaction can decrease oxygen to carbon (O/C) and hydrogen to carbon (H/C) molar ratios, increase the higher heating value, and reduce the transportation cost of biomass [22,23,24]. Thus, the majority of previous studies are focused on the torrefied products, the changes in physicochemical properties of the fuel, and the influence of torrefied biomass on the subsequent pyrolysis under different experiment conditions [25,26,27]. Kanwal et al. reported that torrefaction could enhance the higher heating value of the sugarcane bagasse, especially at 300 °C, in which the HHV of sugarcane bagasse increased from 16.53 MJ/kg to 24.01 MJ/kg [28]. Compared with raw bamboo, torrefied bamboo could decrease the molar ratio of O/C and reduce the content of oxygen-containing functional groups because of the degeneration of volatiles with increasing torrefaction temperature [29]. Cardona et al. reported that the torrefied eucalyptus contained less fiber content, O/C and high HHV at 250 °C–300 °C, compared with the raw eucalyptus [30]. Similarly, after studying the torrefaction of the beechwood chips at different temperatures, Colin et al. also drew the same conclusion [31]. Mei et al. investigated the influence of the torrefaction of different biomass on the oxygen-containing functional groups and found the fiber structure was more significantly cleaved at high temperatures [32]. In addition, Hu et al. investigated the influence of conversion behaviors and the reaction kinetics of the torrefied corn stalk on the pyrolysis and concluded that the torrefied corn stalk could enhance the reactivity of biomass particle in various pyrolysis reactions [33]. Singh et al. studied the characteristics of the bio-oil of the torrefied Acacia nilotica compared with the raw Acacia nilotica in pyrolysis. The result showed that the torrefaction significantly changed the components of the pyrolysis oil, reduced the content of water of the pyrolysis oil, and increased the HHV of the pyrolysis oil [15]. Boateng et al. also obtained a similar conclusion in their studies [34]. Xin et al. reported the torrefaction can increase the yield of H_2_ and CO of pyrolysis gases products compared to the raw biomass, and that the yield of H_2_, CO notably increased with increasing torrefaction temperature in pyrolysis [35]. Though torrefaction is considered as a biomass upgrading technology, many unsolved problems still need to be explored. Achieving consistent and tightly regulated product quality, expanding the process, and densification of the product are the most crucial aspects. Product yield is more important than product quality in continuous torrefaction operations since the reactor must be managed to function at a constant state, making it difficult to assess mass and energy yield directly.

Corn stalk is the most widely distributed commercial crop in China, which is also a potential renewable feedstock. Currently, many researches focus on the effects of torrefaction on the morphology characterization of the bio-char products and the quality of the bio-oil products in pyrolysis. However, studies focusing on the effect on pyrolysis gas products of torrefied corn stalk are rarely reported. In this work, the study on the basic physicochemical properties of corn stalk at different torrefaction temperatures was carried out, and the influence of torrefaction on the microstructure of corn stalk was investigated using XRD, SEM, and FTIR. Moreover, the effect of torrefaction on the pyrolysis of the corn stalk was studied by comparing the pyrolysis products of torrefied corn stalks under different torrefaction temperatures in a vertical fixed-bed reactor.

## 2. Materials and Methods

### 2.1. Biomass Feedstock Preparation

Corn stalk, used as the raw material for the torrefaction experiment, was obtained from northern China. In corn stalk-x (CS-x), x represents torrefaction temperature. The corn stalk was first ground into a powder. Particle size of 30–40 mesh was used for torrefaction. The basic analytic properties of the corn stalk are shown in Table 1.

### 2.2. Experimental Process

The tube furnace reactor system was used as the torrefaction reaction system of corn stalk. The corn stalk was placed in a crystal vessel and transferred to the centre of a quartz tube. Before the torrefaction experiment, the N_2_ of a 99.99% purity was used to purge the reactor for 30 min to remove the air in the tube furnace reactor. Then, the tube furnace reactor system was heated to the target temperature at a fixed heating rate of 10 °C·min^−1^ with an N_2_ flow rate of 50 mL·min^−1^ in the whole experiment. The residence time of the torrefaction experiment was 60 min, and the torrefied corn stalk was collected when completing the experiment. In addition, the different torrefied corn stalks were denoted as CS-X, where X is the torrefaction temperature.

The vertical fixed-bed reactor system was used as the reaction system of the biomass pyrolysis process. The schematic diagram of the fixed-bed reactor used for pyrolysis is shown in Figure 1, which consists of a control unit, a reaction unit, a quartz tube, and a product collection unit. Before the experiment was carried out, pure nitrogen was injected to purge the air inside the quartz tube. The torrefied or raw corn stalk was placed in the quartz cup and transferred to the center of the quartz tube when the temperature was heated to the target temperature. At the same time, the fixed heating rate and the fixed carrier gas of N_2_ were maintained at 50 mL min^−1^ with an experiment time of 40 min. The solid, liquid, and gas products were collected after the experiment was completed.

### 2.3. Analytical Procedures

The proximate analysis, elemental composition, higher heating value (HHV), and the fiber analyses of the CS were determined by the industrial analysis equipment (5E-MAG6600B Shanghai, China), the elemental analysis equipment (Elementar Unicub Munich, Germany), the oxygen bomb calorimeter (5E-AC/PT, Jinan, China), and the fiber analysis equipment (ANKOM A200i Washington, DC, USA) with a Van Soest cellulose content determination method, respectively. The proximate analysis is a general term for the determination of four analytical items including moisture, ash, volatile matter and fixed carbon.

Fourier transform infrared (FTIR) analysis was performed on an FTIR spectrophotometer (Nicolet10 Washington, DC, USA) to identify the distribution of functional groups in CS in the range of 400–4000 cm^−1^.

X-ray diffractograms were acquired by an XRD instrument (XRD-6100 Tokyo, Japan). The CS was laid on an aluminum plate, and the X-ray diffractograms were collected from 5° to 40°(2θ) at a scanning speed of 5°/min. The crystallinity indexes (CrI) of the raw and torrefied corn stalks were calculated using the Sega-l’s method [17] (1):(1)$$CrI(\%)=\frac{\left(I_{002}-I_{am}\right)}{I_{002}}\times 100\%,$$
where *I*_002_ is the maximum peak intensity at 2θ ≈ 22° while Iam is the minimum intensity at 2θ of approximately 18.1°.

The microstructures of CS were examined using field emission scanning electron microscopy (SEM SUPRA 55 Berlin, Germany) to study the morphologies.

## 3. Results and Discussion

### 3.1. Fuel Characteristics of Raw and Torrefied Corn Stalk

The mass yields of the torrefied corn stalk for CS-200, CS-225, CS-250, CS-275 and CS-300 were 96.87%, 93.31%, 84.83%, 71.39% and 54.92%, respectively. This result indicated that the torrefaction temperature had a significant influence on the mass yields of torrefied corn stalk. Table 1 shows the fuel properties of the raw and torrefied corn stalks. The torrefaction temperature significantly impacted the corn stalk. For the proximate analysis, with the torrefaction temperature increased from 200 °C to 300 °C, the content of volatile reduced from 71.44% to 41.79%, while the content of fixed carbon increased from 19.87% to 42.22%. The HHV of raw corn stalk considerably increased from 15.48 MJ/kg to 21.31 MJ/kg when corn stalk was torrefied at 300 °C. The enhancement of HHV is mainly attributed to the decrease in the molar ratio of O/C and the moisture content of torrefied biomass [28].

For the ultimate analysis, the contents of the oxygen element and hydrogen element gradually decreased. Specifically, the content of the carbon element was enriched to 56.16%, while the contents of the oxygen and hydrogen elements decreased to 20.89% and 4.63% when corn stalk was torrefied at 300 °C, respectively. In addition, the same trend was observed in the molar ratios of H/C and O/C of the corn stalk, as shown in Figure 2. The molar ratios of H/C and O/C hardly changed at low temperatures but significantly decreased at high temperatures. In particular, the molar ratios of H/C and O/C significantly decreased to 0.99 and 0.27, respectively, when the torrefaction temperature increased to 300 °C, which is comparable to conventional coals in China such as anthracite, lignite, and peat [3,36]. Overall, the results of the C, H, O and the molar ratios of H/C and O/C can be attributed to the dehydrogenation and deoxygenation reactions in torrefaction [21]. In addition, the dehydrogenation and deoxygenation reactions were also readily affected by temperature.

### 3.2. Fiber and XRD Analysis

The fiber contents of the raw and torrefied corn stalk are shown in Figure 3. The content of lignin increased from 9.22% to 69.88%. In contrast, the contents of hemicellulose and cellulose decreased from 29.38% and 42.04% to 0.33% and 12.33%, respectively, with the torrefaction temperature increased from 200 °C to 300 °C. The variation in fiber content primarily depends on the structural nature. As for cellulose, because it is a linear long-chain carbohydrate polymer and contains a unique crystal structure [28,37], the variation of cellulose also can be investigated by the XRD analysis of the crystallinity index (CrI) of cellulose. Figure 4 presents the XRD analysis of corn stalk, which clearly shows that the CrI hardly changed under 225 °C while intensity decreased from 45.3% to 18.5% when the temperature increased from 250 °C to 300 °C. In addition, Kanwal et al. also reported that the cellulose decomposed at 275–300 °C in torrefaction because of structural liability [28]. Hemicellulose is a branched polymer with a low molecular weight and low polymerization degree [38,39], suggesting that hemicellulose is readily depolymerized at low torrefaction temperatures. In contrast, lignin is a complex phenolic polymer consisting of three phenylpropane monomers and a biopolymer with a three-dimensional network structure [39]. Thus, the lignin exhibits a stable molecular structure and high thermal stability with partial degradation at the higher torrefaction. The increase in lignin occurs mainly due to the formation of insoluble acid products with increasing temperature [40].

### 3.3. FTIR Analysis

The changes in the chemical structure of the torrefied corn stalk and the influence of torrefaction temperature on the chemical functional groups were investigated by using Fourier transform infrared spectroscopy. Figure 5 shows the results of the FTIR analysis of the corn stalks. Various oxygen-containing organic functional groups in fibers such as -OH, C=O, and C-O-C were clearly observed in the infrared spectra.

The absorbance intensity of -OH band (3460 cm^−1^) was slightly reduced at low torrefaction temperatures (<225 °C) but significantly decreased when the temperature increased from 250 °C to 300 °C. This result is attributed to the decomposition of the hydroxy groups of the hemicellulose caused by high temperature and the evaporation of water. The absorbance intensity of the C-H band (2920 cm^−1^) corresponded to the aliphatic –CH_3_ and –CH_2_– of the hemicellulose and cellulose. The absorption intensity exhibited a decreasing trend with increasing torrefaction temperature, especially at higher temperatures (>275 °C). The thermostability of the cellulose impacted the degradation of the aliphatic regions, in which the CH_4_ was most likely released via the cleavage of the C-H bond, according to the report by Ma et al. [38]. The absorbance intensity of the C=O band (1720 cm^−1^) corresponded to the carboxyl and carbonyl groups in the cellulose and hemicellulose. The absorption intensity decreased at 250 °C–300 °C, which was mainly attributed to the enhanced decarboxylation and decarbonylation reactions promoted by increasing temperature. Meanwhile, the cleavage of the C=O band lead to the conversion of aldehydes and ketones into liquid products such as alcohols, ethers, and acids [28]. The absorbance intensity of the C-O and C-H (1106 cm^−1^–989 cm^−1^) was primarily attributed to aliphatic –CH_3_ or phenolic–OH groups in the cellulose, indicating that the complete conversion of methyl and hydroxyl groups during the torrefaction process.

As for the C=C band (1535 cm^−1^–1444 cm^−1^), it corresponded to the aromatic structure of the C=C or benzene ring skeleton of the lignin. However, compared with other above-mentioned chemical functional groups such as O-H and C-H, the absorption intensity of the C=C band only decreased at 300 °C. This result indicates that the degradation of the C=C group of the lignin only occurred at high torrefaction temperatures [37]. Overall, these chemical functional groups of the corn stalk decomposed partially or completely, which confirmed the degradation of the chemical structure with increasing torrefaction temperature.

### 3.4. Morphological Characterization Analysis

The microstructures of the cross-section of raw corn stalk and torrefied corn stalk at different temperatures are shown in Figure 6. Figure 6a,b show the microstructures of the raw and 200 °C torrefied corn stalks, respectively. These microstructures included smooth surface structure, relatively complete fibrous bundle, and tight tubular structure. These results indicate that the decomposition of hemicellulose and cellulose of corn stalk did not occur under low torrefaction temperatures. As reported in other studies, the raw biomass with presence of the fibers also has a smooth surface compared to the torrefied biomass, and it also more difficult to grind into small particles [41]. Figure 6c,d shows that the cracks and pores were generated on the surface, and gaps started to distribute between fibrous bundles with a further increasing torrefaction temperature. These results can be attributed to obviously enhanced dehydroxylation and carbonization reactions and the significant degradation of hemicellulose that is a linker between cellulose and lignin. Meanwhile, these structural changes promoted the release of the volatile products from corn stalk. In Figure 6e, it is apparent that the more visible pores were generated because the increase in temperature promoted the decomposition of cellulose. As the torrefaction temperature increased to a higher degree, as shown in Figure 6f, the fiber structure entirely disappeared, and tiny dispersed particles coalesced into independent tubular structures because of the degradation of cellulose and lignin. Kanwal et al. also reported that the sugarcane bagasse with 300 °C torrefied, the fiber structure disappeared, while the tiny dispersed particles generated. Meanwhile, the release of volatile matter in torrefaction increased the number of pores in the structure with the torrefaction temperature increased [21,28].

### 3.5. Pyrolysis Product Distribution of Torrefied Corn Stalk

Figure 7 shows product distribution of the raw and torrefied corn stalk in pyrolysis. With increasing torrefaction temperature, the yield of solid products of corn stalk increased, while the yield of liquid and gas products of corn stalk gradually decreased. Particularly, when the torrefaction temperature increased from 250 °C to 300 °C, the yield of liquid products and gas products decreased from 35.92% to 23.63% and 30.42% to 21.55%, respectively. The sharp reduction in the yield of liquid and gas products is attributed to the degradation of the volatiles during torrefaction.

### 3.6. Pyrolysis Gas Products of Torrefied Corn Stalk

The influence of torrefied corn stalk on pyrolysis gas products is shown in Figure 8, with the increasing torrefaction temperature. When the raw corn stalk was torrefied at 300 °C, the content of H_2_ increased from 15.53% to 31.82%, whereas the content of CO and CO_2_ decreased from 34.88% and 27.11% to 19.97% and 20.70%, respectively. The torrefaction increased the molar ratio of H_2_/CO to 1.60 when the torrefaction temperature reached 300 °C. The carbon and oxygen functional groups of hemicellulose and cellulose decomposed primarily via the decarbonylation and decarboxylation reactions to release CO_2_ during pyrolysis [38,42]. The methoxyl group (-OCH_3_) of lignin produced massive H_2_ via the decarbonylation reaction during pyrolysis. Therefore, the difference in CO_2_ and H_2_ from raw and torrefied corn stalk is mainly attributed to the variation of hemicellulose, cellulose, and lignin contents in torrefaction. In addition, the total gas yield of torrefied corn stalk at high temperatures was lower than that of raw corn stalk.

Moreover, torrefaction also improved the lower heating value (LHV) of gas products of corn stalk, as clearly observed from Figure 9. As for LHV, it increased at all torrefaction temperatures, while torrefaction hardly influenced the LHV at low temperatures. However, with increasing temperature, especially at 250 °C and 300 °C, the LHV significantly increased to 16.42 MJ/Nm^3^ and 17.83 MJ/Nm^3^, respectively. The increase in LHV was mainly similar to the change in gas products. In other words, the torrefied corn stalk at high temperatures produced an elevated yield of high-calorific-value gas (H_2_ and CH_4_) because of significant variation in the properties of the corn stalk. The increase in torrefaction temperatures promotes the cracking and overwhelming reactions of C=C and C-H in the benzene ring structure and dehydrogenation reactions in the benzene ring structure, thereby increasing the relative content and yield of H_2_. Similarly, the increase in torrefaction temperatures promotes the methamphetamine of the methoxy base of the wood quality to generate CH_4_, thereby increasing the CH_4_ content. Hence, increasing torrefaction temperatures promote lignin decomposition andincrease the thermal value of thermal solution gas.

## 4. Conclusions

This work paves a new strategy for the investigation of the effect of torrefaction on the physiochemical characteristics and pyrolysis of the corn stalk. The results indicated that when the torrefied temperature of corn stalk torrefied reached 300 °C, the H/C and O/C molar ratios of corn stalk decreased to 0.99 and 0.27, respectively, which is close to those of the conventional coals in China such as anthracite, lignite, and peat. The oxygen-containing functional groups of torrefied corn stalk at 300 °C sharply declined, and the C-H and O-H groups were extremely difficult to detect. The fibrous structure of the corn stalk surface was constantly being destroyed, and lots of pores and cracks appeared with increasing torrefaction temperature. The product distribution and gas product quality of the torrefied corn stalk in pyrolysis significantly changed compared with those of the raw corn stalk. The content of H_2_ and the molar ratio of H_2_/CO at 300 °C were prominently enhanced to 31.84% and 1.60 in pyrolysis, respectively. Therefore, torrefaction has an important influence on the improvement of the quality and the subsequent high-value application of the biomass raw material.

## Figures and Tables

**Figure 1 polymers-15-04069-f001:**
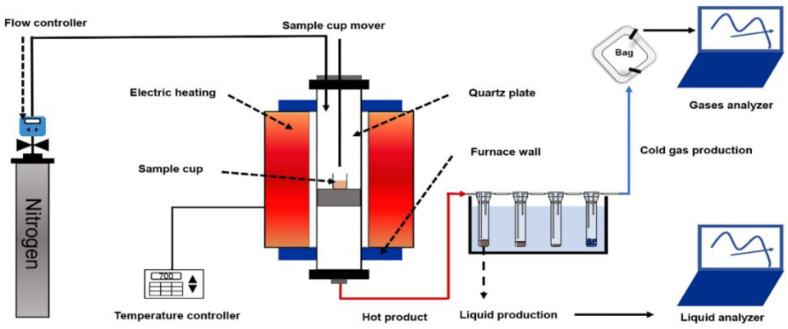
The pyrolysis reaction system of the fixed-bed reactor.

**Figure 2 polymers-15-04069-f002:**
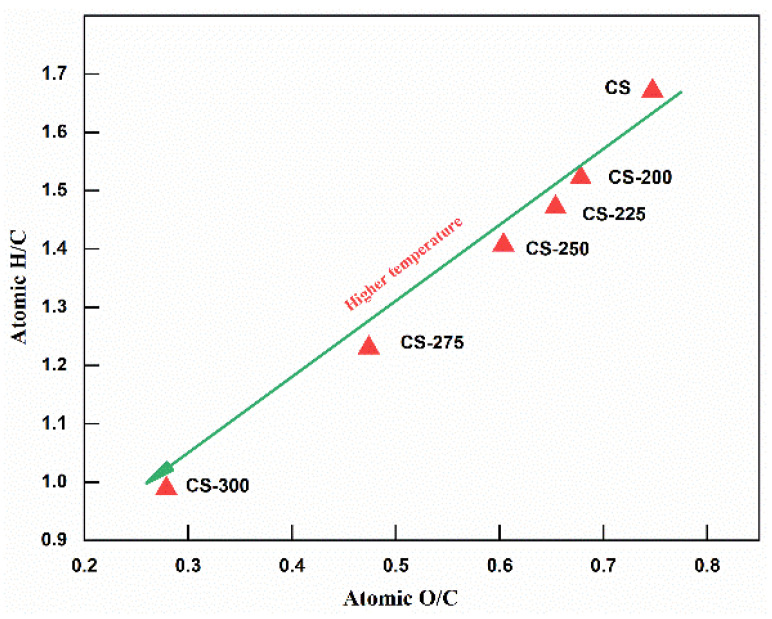
Van Krevelen plot of raw and torrefied corn stalk at different torrefaction temperatures.

**Figure 3 polymers-15-04069-f003:**
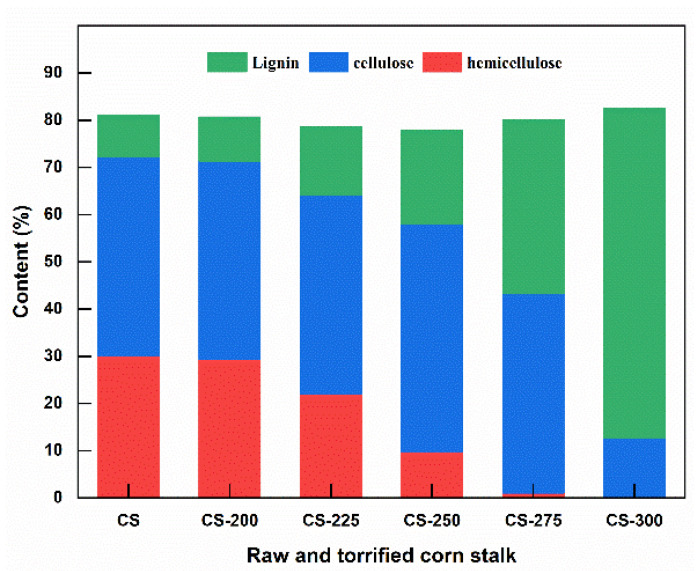
Fiber contents of the raw and torrefied corn stalk at different temperatures.

**Figure 4 polymers-15-04069-f004:**
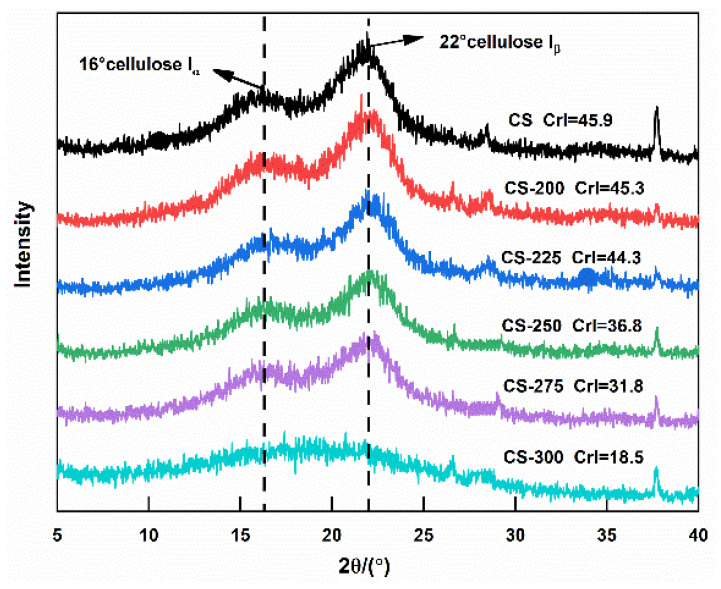
X-ray patterns of torrefied corn stalk at different torrefaction temperatures.

**Figure 5 polymers-15-04069-f005:**
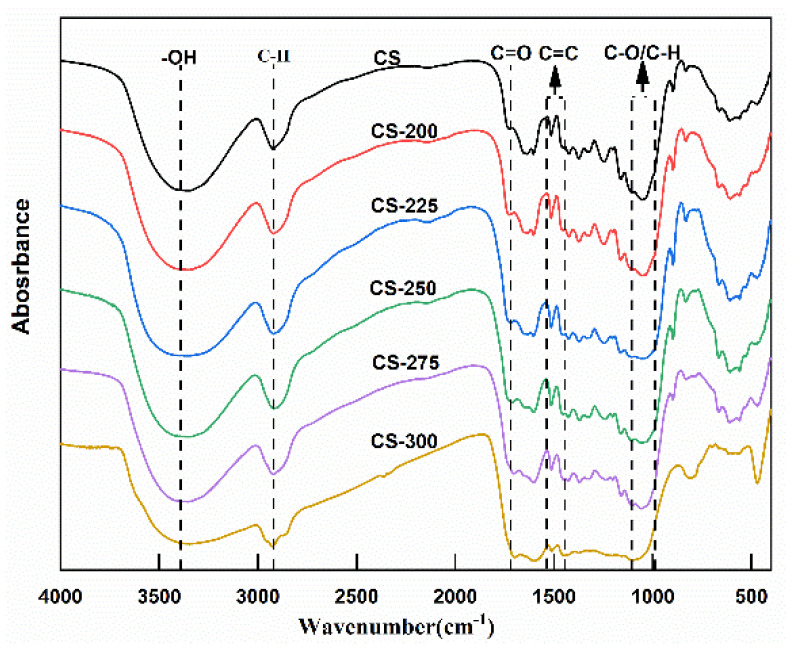
FTIR analysis of torrefied corn stalk at different torrefaction temperatures.

**Figure 6 polymers-15-04069-f006:**
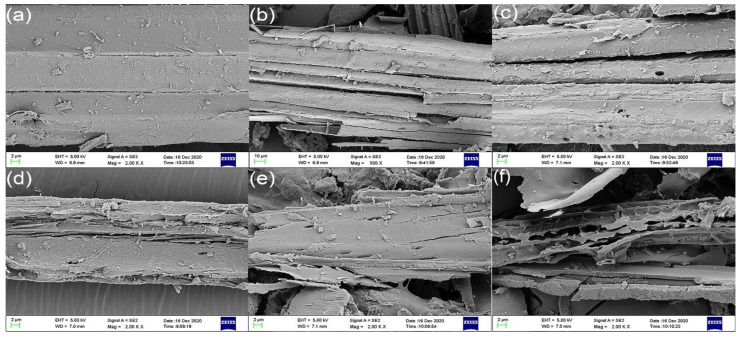
SEM images of corn stalk (**a**) raw, (**b**) torrefied at 200 °C, (**c**) torrefied at 225 °C, (**d**) torrefied at 250 °C, (**e**) torrefied at 275 °C, (**f**) torrefied at 300 °C.

**Figure 7 polymers-15-04069-f007:**
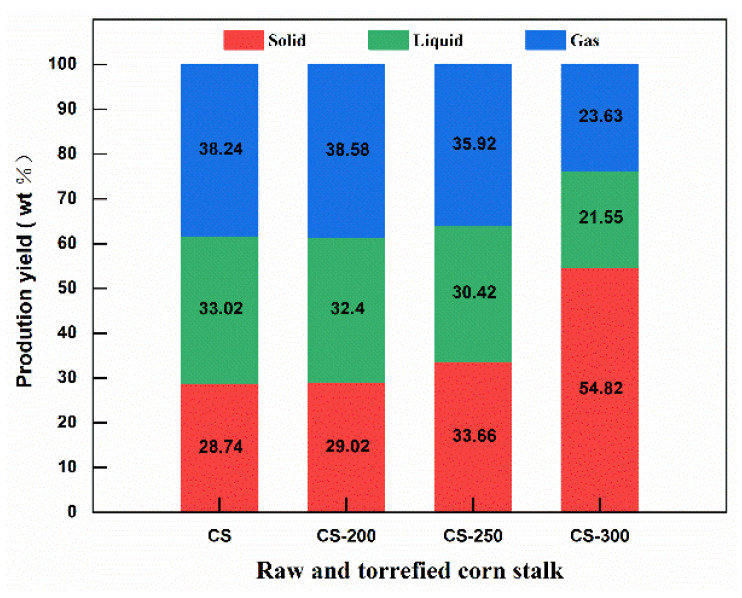
The distribution of pyrolysis products of torrefied corn stalk at 700 °C.

**Figure 8 polymers-15-04069-f008:**
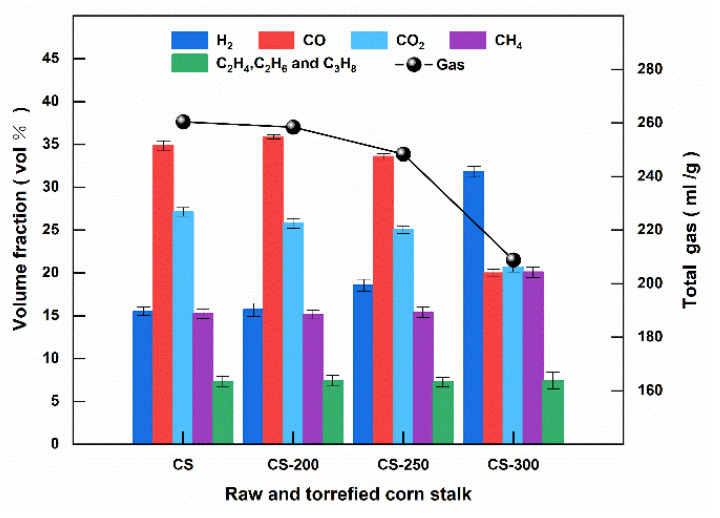
The volume fraction of gas products and the gas yield of torrefied corn stalk in pyrolysis at 700 °C.

**Figure 9 polymers-15-04069-f009:**
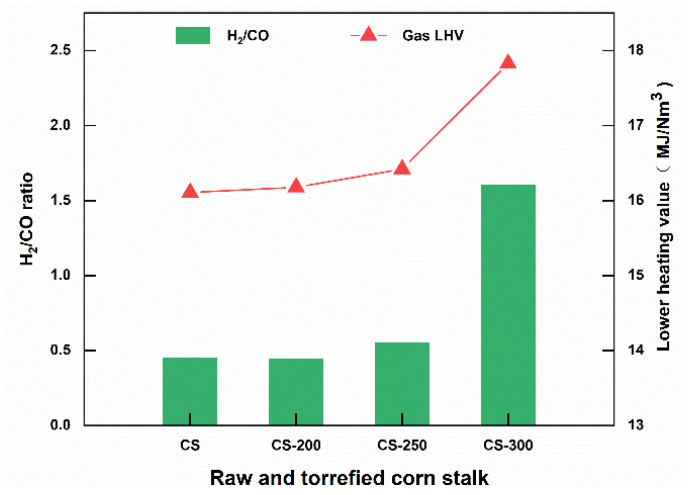
The molar ratio of H_2_/CO and the LHV of pyrolysis gas products of torrefied corn stalk at 700 °C.

**Table 1 polymers-15-04069-t001:** Fuel properties of the raw and torrefied corn stalk.

Sample	VM	FC	Ash	C	H	O	N	S	HHV
CS	71.44	19.87	8.69	41.96	5.84	41.81	1.09	0.60	15.48
CS-200	70.23	20.64	9.13	44.03	5.59	39.81	1.19	0.25	15.58
CS-225	69.11	21.52	9.37	44.77	5.49	39.03	1.10	0.24	15.96
CS-250	65.76	24.09	10.15	45.84	5.37	36.90	1.51	0.23	16.18
CS-275	57.38	30.50	12.12	49.78	5.10	31.45	1.39	0.15	18.31
CS-300	41.79	42.22	15.99	56.16	4.63	20.89	2.09	0.24	21.31

VM = volatile matter, FC = fixed carbon, C = carbon, H = hydrogen, O = oxygen, N = nitrogen, S = sulfur, those data are presented as % of total dry basis, and HHV = higher heating value (MJ/kg).

## Data Availability

Not applicable.

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
