# Peer review of "Effect of Torrefaction on the Physiochemical Characteristics and Pyrolysis of the Corn Stalk"

_polymers, 2023, doi:10.3390/polym15204069_

Round 1
Reviewer 1 Report
1. In the Introduction, unsolved problems must be described.
2. What does "proximate analysis" (line 108) mean? This needs to be explained.
3. What are CS-200, CS-225, CS-250, CS-275 and CS-300 (Table 1)? This needs to be explained.
4. Conclusions are similar to an abstract. New obtained results must be explained in Conclusions. There is no need to describe the obvious facts. "What is happening" and "why it is happening" should be briefly described in Conclusions.
Author Response
Response to reviewers
We are very grateful to receive the editors’ and reviewers’ comments on our manuscript entitled " Effect of torrefaction on the physiochemical characteristics and pyrolysis of the corn stalk" (Manuscript ID: polymers-2615459). We thank the editor or giving us the opportunity to revise our manuscript. We studied the reviewers’ useful and constructive comments very carefully and made corresponding modifications in the revision, which is marked in red color. The changes we made in the revised text and responses to each of the comments organized in a point-by-point fashion are shown as below.
Reviewer #1:
Comment 1: In the Introduction, unsolved problems must be described.
Response: Thank you for your valuable suggestion. According to the reviewer’s suggestion, we have added additional discussion on unsolved problems in the revised manuscript, which are marked in red as below.
Action taken (line 69-74) “Though torrefaction is considered as a biomass upgrading technology, many unsolved problems still need to be explored. Achieving consistent and tightly regulated product quality, expanding the process, and product densification are the most crucial ones. Product yield is more important than product quality in continuous torrefaction operations since the reactor must be managed to function at a constant state, making it difficult to assess mass and energy yield directly.”
Comment 2: What does "proximate analysis" (line 108) mean? This needs to be explained.
Response: Thanks for your advice. According to the reviewer's suggestion, we have supplied a detailed explanation on “proximate analysis” in the revised manuscript, which are marked in red as below.
Action taken (line 113-114) “The proximate analysis is a general term for the determination of four analytical items including moisture, ash, volatile matter and fixed carbon.”
Comment 3: What are CS-200, CS-225, CS-250, CS-275 and CS-300 (Table 1)? This needs to be explained.
Response: Thanks for your advice. According to the reviewer's suggestion, we have supplied a detailed explanation on CS-200, CS-225, CS-250, CS-275 and CS-300 in the revised manuscript, which are marked in red as below.
Action taken (line 87) “Corn stalk-x (CS-x), x represented the torrefaction temperature.”
Comment 4: Conclusions are similar to an abstract. New obtained results must be explained in Conclusions. There is no need to describe the obvious facts. "What is happening" and "why it is happening" should be briefly described in Conclusions.
Response: Thank you for your suggestion. According to the reviewer's suggestion, we have modified the conclusion in the revised manuscript, which are marked in red as below.
Action taken: (Line 257-269): “In this work, the effect of torrefaction at different temperatures on the physicochemical and microstructure properties of torrefied corn stalk. The results indicated that when the torrefied temperature of corn stalk torrefied reached 300 °C, the H/C and O/C molar ratios of corn stalk decreased to 0.99 and 0.27, respectively, which are close to those of the conventional coals in China such as anthracite, lignite, and peat. The oxygen-containing functional groups of torrefied corn stalk at 300 °C sharply declined, and the C-H and O-H groups were extremely difficult to detect. The fibrous structure of the corn stalk surface was constantly being destroyed and appeared lots of the pores and cracks with increasing torrefaction temperature. The product distribution and gas product quality of the torrefied corn stalk in pyrolysis significantly changed compared with those of the raw corn stalk. The content of H2 and the molar ratio of H2/CO at 300 °C were prominently enhanced to 31.84% and 1.60 in pyrolysis, respectively. Therefore, torrefaction has an important influence on the improvement of the quality and the subsequent high-value application of the biomass raw material.”

Reviewer 2 Report
Reviewer’s comments:
Effect of torrefaction on the physiochemical characteristics and pyrolysis of the corn stalk
Authors: Lei Chen, 1Xiangqian Chen, 1 Yuxiao Zhao 1 , Xinping Xie, 1 Shuangxia Yang, 1 Dongliang Hua 1 , Chuanlei Wang 2 4 and Tianjin Li *
Manuscript ID: polymers-2615459
The paper under consideration outlines the torrefaction and pyrolysis processes of corn stalk using tube furnace and fixed bed reactor in this experiment, respectively. The changes in physicochemical properties and morphology features of the torrefied corn stalk were investigated. The primary focus of this experimental study lies in the evaluation of the alterations in physicochemical properties and morphological characteristics of corn stalk following torrefaction. The research outcomes indicate that torrefaction holds the potential to enhance the inherent properties of corn stalk, leading to a substantial improvement in the quality of the gas products generated during pyrolysis. In my assessment, I found the article to be impeccably composed and thoughtfully presented. While I am inclined to endorse the acceptance of this manuscript, I would like to suggest a few minor revisions that, once addressed, would further enhance the overall quality of the paper.
Comment
1. Language needs to be extensively polished.
2. There is no clear information how this work is novel from others work.
3. Please thoroughly check the editorial, subscript, superscript and typographical error. For example, Yuxiao Zhao 1, Xinping Xie, 1 in authors name the positioning of digit 1 mismatching.
4. Why XRD peak Iα and I β are complete vanished with increasing temperature but the peak in 2θ 35-40. Still exist. Why?
5. The FESEM data looks quite interesting. With increasing torrefied temperature from 200 to 300 °C the layer structure is changing drastically. Please explain.
6. The sentence in line 254-256 “The carbon and oxygen functional groups of hemicellulose and cellulose decomposed primarily via the decarbonylation and decarboxylation reactions to release CO2 during pyrolysis Provide some evidence to support this statement by citing an article or providing some results.
7. The statement torrefied corn stalk in high temperatures produced elevated yield of high calorific value gas (H2 and CH4). Please elaborate.
The quality of English is good, but there is still room for improvement.
Author Response
Response to reviewers
We are very grateful to receive the editors’ and reviewers’ comments on our manuscript entitled " Effect of torrefaction on the physiochemical characteristics and pyrolysis of the corn stalk" (Manuscript ID: polymers-2615459). We thank the editor or giving us the opportunity to revise our manuscript. We studied the reviewers’ useful and constructive comments very carefully and made corresponding modifications in the revision, which is marked in red color. The changes we made in the revised text and responses to each of the comments organized in a point-by-point fashion are shown as below.
Reviewer #2:
Comment 1: Language needs to be extensively polished.
Response: Thanks for your suggestion, we have consulted professional editing service (LetPub, www.letpub.com) for polishing. We check the text carefully and the changes in the revision are as follows.
In line 12: “for deriving” was added.
In line 14: “in this experiment” was deleted.
In line 24: “the” was added.
In line 25: “phase” “ratio” was added.
In line 35: “worldwide” was changed into “global”.
In line 36: “countries around the world” was changed into “nations”.
In line 52: “the” was deleted.
In line 57: “studied” was changed into “studying”.
In line 103: “centre” was changed into “center”.
In line 163: “the” was changed into “that”.
In line 172: “the” was added.
In line 192: “the” was added.
In line 206: “And” was changed into “As”.
In line 206: “fibres” was changed into “fibers”.
In line 209: “were” was deleted.
In line 246: “an” was added.
Comment 2: There is no clear information how this work is novel from others work.
Response: Thanks for your advice. According to the reviewer's suggestion, we have emphasized the novelties reported in this paper in the revised manuscript, which are marked in red as below.
Action taken (line 14-15) “the novelty is investigating the changes in physicochemical properties and morphology features of the torrefied corn stalk”.
Action taken (line 26-29) “This work paves a new strategy for the investigation of the effect of torrefaction on the physiochemical characteristics and pyrolysis of the corn stalk, highlighting the application potential in the conversion of biomass.”
Comment 3: Please thoroughly check the editorial, subscript, superscript and typographical error. For example, Yuxiao Zhao 1, Xinping Xie, 1 in authors name the positioning of digit 1 mismatching.
Response: We thank the reviewer for pointing out this mistake. We are sorry for making this mistake. We have corrected it in the revised manuscript.
Comment 4: Why XRD peak Iα and I β are complete vanished with increasing temperature but the peak in 2θ 35-40. Still exist. Why?
Response: Thanks for the good comment. The vanish of XRD peak Iα and I β is attributed to decomposition of cellulose. The XRD peak at 2θ 35-40 may be attributed the peak of inorganic salt (Energy 240 (2022)122483), hence it still existed with increasing temperature.
Comment 5: The FESEM data looks quite interesting. With increasing torrefied temperature from 200 to 300 °C the layer structure is changing drastically. Please explain.
Response: Thanks for the good comment. With the increase of torrefied temperature from 200 to 300 °C, the layer structure destroyed possibly owing to the lower stability of the fiber mesh structure at high temperature. The large amount of semi -cellulose also promotes the decomposition of cellulose. At the same time, the increase in the number of holes is also conducive to the release of volatile products in baking.
Comment 6: The sentence in line 254-256 “The carbon and oxygen functional groups of hemicellulose and cellulose decomposed primarily via the decarbonylation and decarboxylation reactions to release CO2 during pyrolysis Provide some evidence to support this statement by citing an article or providing some results.
Response: Thanks for the good comment. As suggested, we have cited the above helpful literatures (Ref. 43, 44) in the revised manuscript.
Comment 7: The statement torrefied corn stalk in high temperatures produced elevated yield of high calorific value gas (H2 and CH4). Please elaborate.
Response: Thanks for the good comment. According to the reviewer's suggestion, we have supplied the additional discussion on high calorific value gas (H2 and CH4), which are marked in red as below.
Action taken (line 248-255): “The increase of torrefaction temperatures will promote the cracking and overwhelming reactions of C = C and C-H in the benzene ring structure and dehydrogenation reactions in the benzene ring structure, thereby increasing the relative content and yield of H2. Similarly, the increase of torrefaction temperatures will promote the methamphetamine of the methoxy base of the wood quality to generate CH4, thereby increasing the CH4 content. Hence, increasing torrefaction temperatures will promote lignin decomposition and the secondary reaction of thermal solution liquid products to generate CH4 and H2, CH4 and H2 output increased, thereby increasing thermal value of thermal solution gas.”
